# Label Noise Robustness for Domain-Agnostic Fair Corrections via Nearest Neighbors Label Spreading

**Nathan Stromberg**
Arizona State University
nstrombe@asu.edu

**Rohan Ayyagari**
Arizona State University
rayyaga2@asu.edu

**Sanmi Koyejo**
Stanford University
sanmi@cs.stanford.edu

**Richard Nock**
Google Research
richardnock@google.com

**Lalitha Sankar**
Arizona State University
lsankar@asu.edu

## Abstract

Last-layer retraining methods have emerged as an efficient framework for correcting existing base models. Within this framework, several methods have been proposed to deal with correcting models for subgroup fairness with and without group membership information. Importantly, prior work has demonstrated that many methods are susceptible to noisy labels. To this end, we propose a drop-in correction for label noise in last-layer retraining, and demonstrate that it achieves state-of-the-art worst-group accuracy for a broad range of symmetric label noise and across a wide variety of datasets exhibiting spurious correlations. Our proposed approach uses label spreading on a latent nearest neighbors graph and has minimal computational overhead compared to existing methods.

## 1 Introduction

In order to ensure fairness between subpopulations, an important metric to consider is the lowest accuracy amongst all subpopulations, often referred to as worst-group accuracy (WGA). State-of-the-art methods for optimizing WGA utilize group membership information to modify the training loss [1, 12, 21, 20, 22] or the distribution of the training data [10, 6, 11] in order to account for imbalance amongst groups and successfully train a classifier which is fair across groups.

While these methods can achieve astounding WGA, many have significant computational and data costs. Last-layer retraining (LLR) has emerged as a popular method to adapt existing deep models with minimal overhead while ensuring fairness [10, 11, 22, 20]. Additionally, the necessity of group membership information has been relaxed as new methods infer group membership by utilizing auxiliary classifiers to detect minority groups (generally known as two-stage methods) [11, 6, 12, 16, 22]. Taken together, it is now possible to correct powerful deep models for worst-group accuracy with low computational and data cost. Two methods using this approach are SELF [11] and RAD [22].

Unfortunately, class labels are often noisy [25] in practice. Further, Oh et al. [16] demonstrated just how fragile most two-stage model corrections are in the face of this issue. While Oh et al. [16] presented a method with improved robustness to label noise, their method requires fully retraining the base model and cannot match the performance of existing methods at low noise levels. To address these issues, we connect the existing literature on label propagation and worst-group accuracy correction, i.e., we utilize label spreading on the latent nearest neighbors graph to correct for noisy labels and two-stage LLR methods to maximize WGA without domain annotations. We list our main contributions below.

38th Conference on Neural Information Processing Systems (NeurIPS 2024).

- We highlight the failure of state-of-the-art (SOTA) LLR methods, in particular, SELF [11] and RAD [22], under target label noise.

- We introduce an elegant label correction preprocessing method that significantly improves the performance of the SOTA LLR methods under label noise. The key insight here is that LLR models operate on largely separable embeddings, and therefore, label noise can be potentially corrected using label propagation techniques.

- For various spurious correlation datasets, we highlight the effectiveness of our modular approach when combined with both SELF and RAD and compare it to domain-aware and full retraining methods.

## 1.1 Related Work

**Subgroup Robustness** Downsampling has become one of the most common methods of increasing WGA. Kirichenko et al. [10] propose deep feature reweighting (DFR), which downsamples majority groups to the size of the smallest group to rebalance the retraining set. Chaudhuri et al. [4] explore the effects of downsampling theoretically and show that downsampling can increase WGA under certain data distribution assumptions. LaBonte et al. [11] propose using class-balancing when group membership information is unavailable and demonstrate its efficacy.

Upweighting is a popular alternative to downsampling as it utilizes all available data. Idrissi et al. [7] show that upweighting relative to the proportion of groups can achieve strong WGA, and Welfert et al. [26] prove downsampling and this form of upweighting are statistically equivalent. Upweighting has been extended to domain annotation-free settings by Qiu et al. [20] through loss-weighted learning.

Domain annotation-free methods often use a secondary model to identify minority groups. These are referred to as two-stage methods. Qiu et al. [20] use the pretrained model itself but do not explicitly identify minority examples and instead upweight proportionally to the loss. Unfortunately, this ties their identification method to the choice of the loss. Liu et al. [12] consider fully retraining the pretrained model as opposed to only the last layer, but their method of minority identification using an early stopped model is considered in the last layer in LaBonte et al. [11]. LaBonte et al. [11] not only consider early stopping as implicit regularization for their identification model, but also dropout (randomly dropping weights during training). Stromberg et al. [22] consider an explicit $\ell_1$ regularizer, building on the work of LaBonte et al. [11] and Kirichenko et al. [10].

Wei et al. [25] show that human annotation of image class labels can be noisy with up to a 40% noise proportion, thus motivating the need for robust methods for WGA. Oh et al. [16] consider robustness to class label noise. Their method, END, utilizes predictive uncertainty from a robust identification model to select an unbiased retraining set. Their method struggles against SOTA methods at low noise levels and is not easily adapted to other error set-selection techniques. Still, it achieves strong performance in the high noise regime where other two-stage full retraining methods fall apart.

**Label Propagation** The problem of spreading labels from a small, but well-annotated set of reference points to a larger unlabeled set has been extensively explored in the literature. For a survey of basic methods of label propagation on a graph, see Bengio et al. [2] and references therein. The graph on which labels are propagated is critical to the success of these methods. Common graphs are the $k$-nearest neighbors (kNN) graph, the adjacency matrix of an undirected graph, or an underlying directed acyclic graph (DAG).

When labels are noisy, the objective of label propagation may be ill-equipped to correct these labels, but the general principles can be of use. For a restricted classifier setting, Gao et al. [5] prove the robustness of the kNN learning procedure to label noise and give a sense of how to choose $k$, the number of nearest neighbors, via an excess risk bound. This inspires our empirical analysis and our results verify the trends suggested by their bound. Iscen et al. [9] utilize neighbor consistency to deal with noisy labels when training deep models and frame their method as an implicit label propagation. Patrini et al. [19] consider label noise and possible corrections via loss and logit reweighting.

Label propagation and spreading have also been used in the semi-supervised learning setting for both general models and those focused on subgroup fairness. Iscen et al. [8] include label propagation in order to learn general deep models in a semi-supervised manner. Nam et al. [15] utilize an auxiliary model inspired by the label propagation objective to predict group labels on unlabeled samples. This

allows for semi-supervised fair training. Unfortunately, their method is very costly as it involves training both a spurious attribute prediction model and running GroupDRO [21].

**Our work** combines the ideas of two-stage last-layer corrections of LaBonte et al. [11], Stromberg et al. [22] with simple label propagation techniques in the latent space of the original (unfair) model. Because we have a low-dimensional embedding in which clean examples are well-separated by a linear classifier, we can utilize kNN label spreading with many nearest neighbors, which is enough to correct most label noise in the training data. With this preprocessing method, we can then proceed with any two-stage last-layer WGA correction to achieve fairness without domain information. Thus, our method is an elegant plug-and-play addition to last-layer subgroup robustness methods, which achieves SOTA test WGA when training with noisy class labels.

## 2 Problem Setup

Let $\mathcal{D} = \{(X_i, Y_i)\}_{i=1}^n$ be a dataset of $n$ iid examples with features $X_i \in \mathcal{X}$ and class label $Y_i \in \mathcal{Y}$. Additionally let $\mathcal{G} = \{\mathcal{G}_i\}_{i=1}^k$ be a partition of $\mathcal{D}$, that is each $\mathcal{G}_i \subseteq \mathcal{D}$, corresponding to the $k$ groups of interest. In general, this partition is unknown at training time.

We are given a pre-trained deep model

$$f(x; \theta) = \arg \max_{y \in \mathcal{Y}} \sigma_y(\langle \Phi(x), \theta \rangle), \tag{1}$$

where $\Phi : \mathcal{X} \to \mathbb{R}^d$ is an embedding function and $\sigma : \mathbb{R} \to [0, 1]^{|\mathcal{Y}|}$ such that $\sigma_y$ is the estimate of the probability that $X$ is in class $y$. An example of $\sigma$ is the oft-used softmax function. We assume that $f$ is trained on clean data. While this may seem like a strong assumption, it can be relaxed as recent work suggests that feature embeddings are typically robust to label noise [9, 17].

We seek to retrain solely the last layer of $f$ in order to increase the worst-group accuracy of the model. That is we learn a $\theta_{\text{WGA}}$ such that

$$\theta_{\text{WGA}} = \max_{\theta} \min_{\mathcal{G}_i \in \mathcal{G}} \frac{1}{|\mathcal{G}_i|} \sum_{(x,y) \in \mathcal{G}_i} \mathbb{1}(f(x; \theta) = y). \tag{2}$$

That is, $\theta_{\text{WGA}}$ maximizes the worst-group accuracy (WGA). We will use WGA as our metric for all of our discussions moving forward.

### 2.1 Label Noise Model

We assume that we observe not $\mathcal{D}$, but $\tilde{\mathcal{D}} = \{(X_i, \tilde{Y}_i)\}_{i=1}^n$ where $Y$ has been affected by symmetric label noise with noise level $p \in (0, 1/2)$. In general, this noise could be caused by human error, data corruption, or malevolence. Patrini et al. [19] consider a wider class of class-dependent noises and show that in general, predictors can be made robust to such noise with relatively simple adjustments to the loss function or data distributions. This simple correction is not helpful in two-stage methods as demonstrated by Oh et al. [16]; we expand on this further in Section 2.3.

### 2.2 Basic Last-Layer Model Corrections for WGA

When we have information about the group membership $\mathcal{G}$, standard methods are to upweight examples proportional to their group size or to remove samples from larger groups. Specifically, group upweighting (GUW) seeks to minimize the following objective:

$$\theta^{(GUW)} = \max_{\theta} \frac{1}{k} \sum_{\mathcal{G}_i \in \mathcal{G}} \frac{n}{|\mathcal{G}_i|} \sum_{(x,y) \in \mathcal{G}_i} \mathbb{1}(f(x; \theta) = y), \tag{3}$$

where the inner sum is upweighted by a factor inversely proportional to its prevalence in the dataset. Group downsampling (GDS) takes a similar approach, but sub-samples all groups which are larger than the smallest group. Denoting $n_{\min} = \min_{\mathcal{G}_i \in \mathcal{G}} |\mathcal{G}_i|$ as the size of the smallest group, we can write the GDS objective as:

$$\theta^{(GDS)} = \max_{\theta} \frac{1}{k n_{\min}} \sum_{\mathcal{G}_i \in \mathcal{G}} \sum_{(x,y) \in \overline{\mathcal{G}_i}} \mathbb{1}(f(x; \theta) = y), \tag{4}$$

where $\overline{\mathcal{G}_i}$ is a downsampled version of $\mathcal{G}_i$ with size $n_{\min}$.

These two approaches are examined theoretically in Chaudhuri et al. [4], Welfert et al. [26], though their performance in the presence of label noise is unclear. Patrini et al. [19] suggest that a related upweighting factor to Equation (3) may be helpful in combating label noise (the so-called backwards correction) suggesting that GUW may be robust to label noise. Welfert et al. [26] prove that GUW and GDS are equivalent in the statistical setting (infinitely many samples), which suggests that GDS should also have some resilience to label noise. We explore this empirically in Section 4.

## 2.3 Two-Stage Last-Layer Model Correction

When group membership information is unavailable, techniques that build on GUW or GDS have been explored. Broadly they fall into the category of two-stage corrections [6, 12, 16, 14], and more specifically in our setting, last-layer model corrections for worst-group accuracy [11, 22]. These methods make the assumption that the groups of interest are determined by a tuple of class label $Y$ and domain label $D$. Additionally there is a spurious correlation between $Y$ and $D$ which causes the base model $f$ to perform poorly on minority groups. This same correlation is exploited when correcting the model in that an error set, $\mathcal{E}$, is constructed that is likely to contain minority examples:

$$\mathcal{E} = \{(X, Y) \in \mathcal{D} : f_{\text{bias}}(X) \neq Y\}, \tag{5}$$

where $f_{\text{bias}}$ is the base model $f$ for LaBonte et al. [11] and a highly regularized model intended to increase performance in Stromberg et al. [22]. This error set is then used as a proxy for a group-balanced or minority-dominated subset, respectively, for fair retraining. We present a simplified version of the RAD [22] and SELF [11] algorithms in Algorithm 1 and Algorithm 2 respectively. Note that in this work we focus on the misclassification version of SELF, but other variations are discussed by LaBonte et al. [11]. The intuition behind these methods is that the error set will likely contain mostly minority examples as the majority should be easily captured by the biased classifier.

The construction of this error set is precisely why using a noise-robust loss [16] or a corrected loss [19] is not enough to achieve robustness to label noise for two-stage LLR methods. The robust models will still (correctly) misclassify noisy examples, including them in the error set for retraining (see Appendix C for experimental results in this setting). This results in an error set which is dominated by noisy majority points rather than clean minority points. This is exacerbated by SELF which keeps only the examples with highest loss (most likely to be noisy). We present a proof of this failure for RAD in a group-conditional Gaussian setting in Appendix B Proposition 2. Oh et al. [16] circumvent this problem by changing the construction of the error set to depend on the entropy of the predictions rather than their correctness. While this increases robustness to label noise, it comes at the cost of clean performance. We instead focus on cleaning the labels so that we can realize the performance gains of existing two-stage methods.

---

**Algorithm 1** RAD [22] Algorithm

**Input:** $\mathcal{D} = (x_i, y_i)$ for $i \in [n]$, $c$: penalty factor, $\lambda$: upweight factor
$\mathcal{E} \leftarrow \emptyset$
Learn $\theta_{\text{bias}}$ minimizing

$$\sum_{(x,y) \in \mathcal{D}} \ell(x, y; \theta) + c\|\theta\|_1 \tag{6}$$

$\hat{\boldsymbol{y}} \leftarrow f(\boldsymbol{x}; \theta_{\text{bias}})$
**if** $\hat{y}_i \neq y_i$ **then**
  $\mathcal{E} \leftarrow \mathcal{E} \cup (x_i, y_i)$
**end if**
Learn $\theta_{\text{RAD}}$ minimizing

$$\sum_{(x,y) \in \mathcal{D} \setminus \mathcal{E}} \ell(x, y; \theta) + \lambda \sum_{(x,y) \in \mathcal{E}} \ell(x, y; \theta) \tag{7}$$

**Return:** $\theta_{\text{RAD}}$

---

**Algorithm 2** SELF [11] Algorithm

**Input:** $\mathcal{D} = (x_i, y_i)$ for $i \in [n]$, $n_{\text{sub}}$: size of reweighting set
$\mathcal{E} \leftarrow \emptyset$
$\hat{\boldsymbol{y}} \leftarrow f(\boldsymbol{x}; \theta)$
**if** $\hat{y}_i \neq y_i$ **then**
  $\mathcal{E} \leftarrow \mathcal{E} \cup (x_i, y_i)$
**end if**
Subsample $\mathcal{E}$ to select $n_{\text{sub}}$ examples with highest loss
Subsample $\mathcal{E}$ so that each class is equally represented
Learn $\theta_{\text{SELF}}$ minimizing, starting from $\theta$

$$\sum_{(x,y) \in \mathcal{E}} \ell(x, y; \theta) \tag{8}$$

**Return:** $\theta_{\text{SELF}}$

---

# 3 Label Spreading for Robust Worst-Group Accuracy

To correct for label noise in the training data, we will consider the labels of each point's nearest neighbors. We denote the $k$ nearest neighbors matrix as $V_k$ where $V_k(i, j)$ is 1 if $X_j$ is one of the $k$ nearest neighbors of $X_i$ in $\ell_2$ distance and 0 otherwise. Thus $\frac{1}{k} V_k$ is a row-stochastic matrix with a uniform distribution on each point's $k$ nearest neighbors. Note that $V_k$ is not necessarily symmetric and so defines a directed graph.

We update the estimated labels iteratively by taking a uniformly-weighted majority vote of each point's nearest neighbors. The idea is that noise is added at random to each class, thus it is likely that for noise level $p$, $1 - p$ proportion of a given query point's neighbors have clean labels. It is intuitive to select a higher $k$ for increasing $p$ in order to minimize the effect of local noise density. Thus most noisy points will be corrected after just one round of spreading, and noise will continue to diminish unless there is a small cluster which is especially noisy. See Gao et al. [5] for a more complete theoretical treatment of kNN classification with label noise.

To better understand how the choice of $k$ is affected by the noise parameter $p$ consider the following:

**Proposition 1** (Theorem 2 from Gao et al. [5]). *For $k \geq 8$ and symmetric label noise level $p$*

$$\mathbb{E}_{\mathcal{D}}[\mathcal{R}_k] \leq \mathcal{R}^* + \frac{2\mathcal{R}^*}{\sqrt{k}} + \frac{p}{(1 - 2p)\sqrt{k}} + 5 \max\{L, \sqrt{L}\}\sqrt{d} \left(\frac{k}{n}\right)^{1/(1+d)} \tag{9}$$

*where $\mathcal{R}^*$ is the Bayes optimal risk, $\mathcal{R}_k$ is the risk of kNN, $d$ is the data feature dimensions, and $L$ is the Lipschitz constant of the Bayes optimal classifier.*

For $d \gg 1$, minimizing the upper bound in Equation (9) leads to the dependence of $k$ on $p$ as

$$\mathcal{O}\left(\left(\mathcal{R}^* + \frac{p}{1 - 2p}\right)^2\right), \tag{10}$$

corroborating the intuitive choice of larger $k$ for larger amounts of noise.

## 3.1 A Note on Domain Label Spreading

Because of the effectiveness of simple domain-aware LLR methods like GDS and GUW, it may be tempting to consider kNN label spreading for domain noise as well as target noise. In this way, we could correct for the failures of GUW and GDS seen by Stromberg et al. [22] which lead them to propose RAD. The effectiveness of kNN label spreading for target labels is mainly due to the fact that the clean embeddings are close to linearly separable (that is the original base model had high accuracy). In practice, not only are classes linearly separable in the latent space, but are tightly clustered. This has been observed empirically and partially explained theoretically under the framework of neural collapse [18]. Such collapse has not been seen for subgroups when this information is not explicitly utilized in the training of the base model. Thus it is unreasonable to expect that domains should be as well separated as classes. This limits the effectiveness of kNN label spreading for domain labels, but as we demonstrate in Section 4, kNN-RAD can achieve SOTA WGA without using domain labels at all in the training phase. Nam et al. [15] bypass this issue by training a separate neural network to predict group membership using semi-supervised data, effectively causing neural collapse for groups rather than classes. However, their method suffers from poor computational performance and struggles to deal with label noise as demonstrated by Oh et al. [16].

## 3.2 kNN-RAD and kNN-SELF

In order to retrain the base model, we first perform $T$ rounds of kNN label spreading (in practice we take $T = 1$) and then pass the cleaned labels to any last layer method. We denote the combination of these procedures **kNN-RAD**, Algorithm 3, or **kNN-SELF**, Algorithm 4, depending on the two-stage method used after label spreading. While we focus on using embeddings from a model trained on

clean data, prior work suggests that the embeddings should be relatively robust to label noise [9, 17].

---

**Algorithm 3** kNN-RAD Algorithm

**Input:** $\mathcal{D} = (x_i, \tilde{y}_i)$ for $i \in [n]$, $k$: number of neighbors, $c$: penalty factor, $\lambda$: upweight factor

*Calculate* kNN graph $V_k$
$\hat{\boldsymbol{y}}^{(0)} \leftarrow \tilde{\boldsymbol{y}}$
**for** $t = 1, \ldots, T$ **do**
$\quad \hat{\boldsymbol{y}}^{(t)} \leftarrow \mathbb{1}(V_k \hat{\boldsymbol{y}}^{(t-1)} \geq \frac{k}{2})$
**end for**
$\theta_{\text{RAD}} \leftarrow \text{RAD}(\boldsymbol{x}, \hat{\boldsymbol{y}}^{(T)}, c, \lambda)$ (Algorithm 1)
**Return:** $\theta_{\text{RAD}}$

**Algorithm 4** kNN-SELF Algorithm

**Input:** $\mathcal{D} = (x_i, \tilde{y}_i)$ for $i \in [n]$, $k$: number of neighbors, $n_{\text{sub}}$ size of reweighting set

*Calculate* kNN graph $V_k$
$\hat{\boldsymbol{y}}^{(0)} \leftarrow \tilde{\boldsymbol{y}}$
**for** $t = 1, \ldots, T$ **do**
$\quad \hat{\boldsymbol{y}}^{(t)} \leftarrow \mathbb{1}(V_k \hat{\boldsymbol{y}}^{(t-1)} \geq \frac{k}{2})$
**end for**
$\theta_{\text{SELF}} \leftarrow \text{SELF}(\boldsymbol{x}, \hat{\boldsymbol{y}}^{(T)}, n_{\text{sub}})$ (Algorithm 2)
**Return:** $\theta_{\text{SELF}}$

---

## 4 Experiments

We present worst-group accuracies for several representative methods across four large publicly available datasets. Specifically, we compare kNN-RAD and kNN-SELF to END [16] which aims to provide robustness to label noise for two-stage full retraining methods. As baselines we include group upweighting (GUW) and group downsampling (GDS) which are simple and effective, but require domain annotations which are not available to other methods. Additionally, we demonstrate that uncorrected RAD [22] and SELF [11] suffer large performance degradation under label noise.

### 4.1 Experimental Details

We perform our experiments with several common datasets in the literature of worst-group accuracy, including three vision datasets and one text dataset. Following prior work [10, 11], we use half of the validation as retraining data (with noisy target labels) and half as a *clean* holdout. More details can be found in Appendix A.

**CMNIST** [1] is a variant of the MNIST handwritten digit dataset in which digits 0-4 are labeled $y = 0$ and digits 5-9 are labeled $y = 1$. Further, 90% of digits labeled $y = 0$ are colored green and 10% are colored red. The reverse is true for those labeled $y = 1$. Thus, we can view color as a domain, and we can see that the color of the digit and its label are correlated.

**CelebA** [13] is a dataset of celebrity faces. For this data, we predict hair color as either blonde ($y = 1$) or non-blonde ($y = 0$) and use gender, either male ($d = 1$) or female ($d = 0$), as the domain label. There is a correlation in the dataset between hair color and gender because of the prevalence of blonde female celebrities.

**Waterbirds** [21] is a semi-synthetic dataset which places images of land birds ($y = 1$) or sea birds ($y = 0$) on land ($d = 1$) or sea ($d = 0$) backgrounds. There is a correlation between background and the type of bird in the training data but this correlation is removed in the validation data.

**CivilComments** [3] is a text corpus dataset of public comments on news websites. Comments are labeled either as toxic ($y = 1$) or civil ($y = 0$) and the spurious attribute is the presence ($d = 1$) or absence ($d = 0$) of a minority identifier (e.g. LGBTQ, race, gender). There is a strong class imbalance (most comments are civil), though the domain imbalance is modest.

### 4.2 Empirical Evidence for Label Spreading

To empirically evaluate the effectiveness of kNN label spreading, we add symmetric label noise to embeddings from three image datasets and then perform label spreading over 10 rounds. We see in Figure 1 that kNN label spreading can correct most of the noise present in the training data with only one round of spreading. This is likely due to the well-separated nature of the embeddings. CMNIST suffers from decreasing performance with too many rounds or neighbors as the data is not perfectly clustered into classes as it is in the other image datasets. Here, it is likely that the nearest neighbors for the small clusters to the side in Figure 2 are of the opposite class, allowing noisy points to spread more easily than clean points. For this reason, we employ a smaller value of $k$ for CMNIST than for other

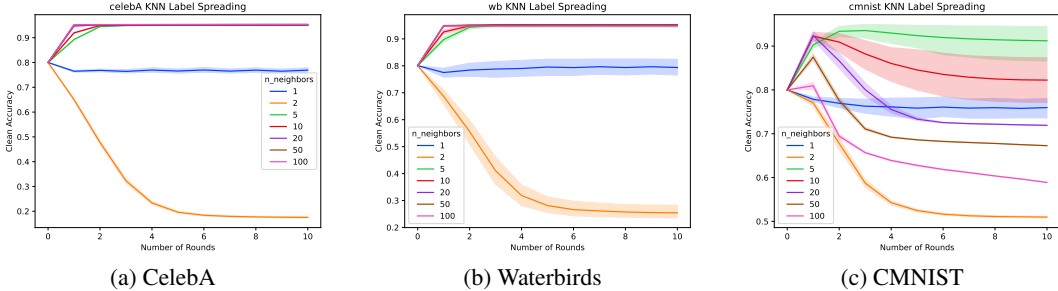

| (a) CelebA | (b) Waterbirds | (c) CMNIST |

Figure 1: Accuracy (and 95% confidence intervals over 10 runs) of predicted labels from kNN under 20% symmetric label noise. CelebA and Waterbirds achieve strong performance with a large number of nearest neighbors, but CMNIST struggles as the number of neighbors or rounds grows too large.

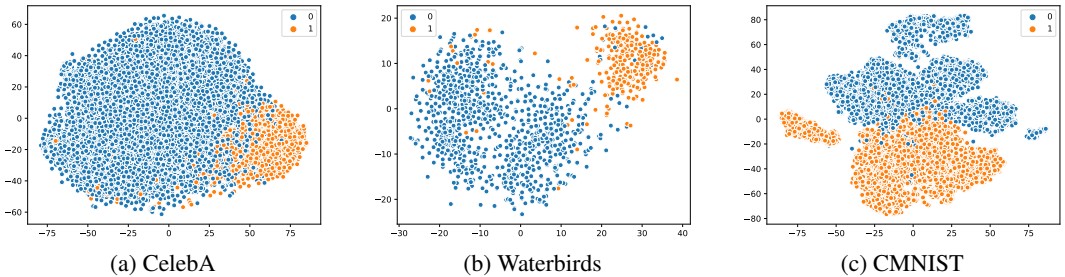

| (a) CelebA | (b) Waterbirds | (c) CMNIST |

Figure 2: tSNE projection of the 2048 dimensional latent embeddings into a 2 dimensional space for visualization. We see that CelebA and Waterbirds show clear class separation while CMNIST has more hierarchical clustering. This could lead to decreased performance of label spreading.

datasets. Distance-weighted voting could be another way to combat this issue. Additionally, because the label spreading process is so quick, we spread for only 1 round in practice. A distance-weighted approach may require more rounds for clean labels to spread throughout the dataset.

### 4.3 Robustness of kNN to Noisy Embeddings

While we assume that the embedding model $\Phi$ is train on clean (but imbalanced) data, it is not always the case in practice. While Iscen et al. [9] suggest that embeddings are relatively robust to noise, they find that the classification head is not. Both RAD and SELF rely on the embeddings and SELF on the classification head as well. Additionally, the kNN label spreading proposed in this work assumes clean embeddings with only symmetric label noise.

To explore how robust our method is to violation of these assumptions, we train base models for both CelebA and Waterbirds on data which contained 20% SLN. We then test the finetuning procedure with varying levels of label noise. We see in Table 1 that both RAD and SELF suffer when using noisy embeddings, and this is only exacerbated by noise in the finetuning set. Still, kNN label cleaning is able to provide a significant boost in performance despite having relatively low-quality embeddings. We see a similar trend for Waterbirds in Table 2 which shows even stronger performance gains for kNN over vanilla two-stage methods.

It is also notable that SELF has significantly worse performance than RAD when using noisy embeddings, likely because SELF additionally reuses the final classification layer of the base model. Previous work [8] has shown that most of the drop in performance when training deep models with noisy data comes from a poor classification head.

### 4.4 Results

We report the worst-group accuracy of each approach and its standard deviation over 10 noise seeds. Note that for END [16], we report the results of Oh et al. [16]. For SELF [11] and RAD [22], we implement their algorithms ourselves to ensure a fair comparison. Each table involves the following:

Table 1: **CelebA WGA (std. dev.)** using embeddings from a noisy base model. We see that kNN label correction still offers a significant increase in performance over vanilla two-stage methods.

| Method | 0% | 10% | 20% | 30% |
|---|---|---|---|---|
| RAD | 80.55 (0.02) | 74.78 (2.32) | 0 (0) | 0 (0) |
| SELF | 31.86 (2.32) | 40.55 (0.66) | 17.68 (16.71) | 50.55 (1.36) |
| KNN - RAD | 80.55 (0.02) | 77.38 (1.7) | 76.56 (2.24) | 69.94 (3.07) |
| KNN - SELF | 31.01 (0.42) | 41.9 (3.84) | 44.8 (4.64) | 46 (7.20) |

Table 2: **Waterbirds WGA (std. dev.)** using embeddings from a noisy base model. We see that kNN strongly outperforms vanilla two-stage methods at every noise level.

| Method | 0% | 10% | 20% | 30% |
|---|---|---|---|---|
| RAD | 86.12 (0.05) | 52.49 (2) | 32.26 (2.99) | 16.94 (2.5) |
| SELF | 67.43 (4.48) | 21.67 (3.12) | 16.71 (1.30) | 6.43 (19.30) |
| KNN - RAD | 84.04 (0.04) | 74.28 (2.63) | 73.78 (3.29) | 61.07 (9.68) |
| KNN - SELF | 68.96 (3.79) | 68.02 (1.9) | 74.90 (2.73) | 49.41 (9.89) |

- Each table is broken into three parts: the first are simple baselines with access to *clean* domain annotations, the second are SOTA two-stage LLR methods for WGA, and the third are robust two-stage methods including our own (last layer retraining) kNN-RAD and kNN-SELF.

- We write GUW* and GDS* to denote the worst-group accuracies of upweighting and downsampling, respectively, that can be achieved with oracle access to *clean* domain labels at training time. Note that such access is *not available* to the two-stage methods.

- We highlight in **bold** both the best domain annotation-free method for each noise level and the method within one standard deviation of the best.

- The "Domain Annotation" column denotes whether a method requires access to domain annotation only at validation (model selection) time or over both training and validation phases. The "Layer" column denotes whether a given method retrains only the *last* layer or the *full* model.

For the CMNIST dataset, in Table 3 we see that kNN label spreading gives small gains over vanilla RAD, but even vanilla RAD is not significantly diminishing in performance with increasing noise levels. This is likely due to the ease of the dataset overall. As noted in Section 3, CMNIST is not as strongly clustered as other image datasets, so it is necessary to use a smaller number of nearest neighbors. Here, kNN-SELF provides small gains over vanilla SELF as both are fairly robust.

For the CelebA dataset, Table 4 shows that kNN-RAD achieves dramatically improved performance relative to vanilla RAD and also over END. Although it retrains only the last layer, kNN-RAD achieves SOTA performance for domain annotation-free methods at every noise level for this dataset. In fact, kNN-RAD is competitive with domain-aware methods such as GDS and GUW across noise levels. Here kNN-SELF falls behind kNN-RAD and END, but still achieves significantly more robustness than vanilla SELF. Note the relatively higher variance of WGA for SELF and kNN-SELF; this is likely due to the downsampling step that induces different data distributions for different seeds.

For the Waterbirds dataset, we see in Table 5 that kNN-RAD strongly outperforms END and is competitive with domain-aware methods across noise levels, improving the performance of even the oracle methods GDS and GUW. Vanilla RAD experiences a catastrophic failure at 20% label noise and above which kNN-RAD is able to correct. SELF performs well at 0% label noise, but quickly degrades with label noise. kNN-SELF achieves dramatic gains over vanilla SELF and maintains an edge over END at every noise level.

For the CivilComments dataset, we see in Table 6 that kNN-RAD is able to match the performance of the domain-aware methods without having access to domain information. Additionally, kNN label spreading demonstrates significant gains over vanilla RAD. SELF struggles on CivilComments even in the experiments of LaBonte et al. [11], so it is no surprise that it performs poorly here. The heavy class imbalance is likely the culprit combined with the class balancing performed in Algorithm 2.

Table 3: **CMNIST WGA (std. dev.)**: GUW* and GDS* denote the worst-group accuracies of upweighting and downsampling, respectively, achieved with oracle access to *clean* domain labels which aren't available to the two-stage methods. We list both the best domain annotation-free method for each noise level and the method within one standard deviation of the best in **bold**. We see that CMNIST is a relatively easy dataset in general, so label noise does not cause dramatic performance loss. Yet, kNN provides some additional robustness for both RAD and SELF. CMNIST is not considered in [16] and thus, results for END are not reported.

| Method | Domain Annotation | Layer | Label Noise (%) | | | |
|---|---|---|---|---|---|---|
| | | | 0 | 10 | 20 | 30 |
| GUW* | Training | Last | 95.27 (0.07) | 95.17 (0.53) | 93.11 (0.94) | 92.05 (1.23) |
| GDS* | Training | Last | 95.37 (0.22) | 94.19(1.02) | 94.35 (1.29) | 93.31 (0.98) |
| RAD [22] | Val | Last | **93.41** (0.79) | 89.62 (0.72) | 89.52 (0.65) | 88.78 (0.89) |
| SELF [11] | Val | Last | 92.04 (0.20) | 89.98 (1.16) | 88.05 (2.35) | **90.58** (1.12) |
| END [16] | Val | Full | - | - | - | - |
| kNN-RAD | Val | Last | **93.42** (0.63) | **92.46** (0.92) | **91.95** (0.86) | **90.50** (3.59) |
| kNN-SELF | Val | Last | 91.77 (0.21) | **92.81** (0.88) | 91.06 (1.06) | **90.16** (3.67) |

Table 4: **CelebA WGA (std. dev.)**: We see that RAD and SELF achieve strong performance at 0% noise, but are not robust at larger noise levels. Here kNN-RAD and kNN-SELF maintain strong performance relative to their vanilla counterparts up to 30% noise and kNN-RAD strongly outperforms END at all noise levels.

| Method | Domain Annotation | Layer | Label Noise (%) | | | |
|---|---|---|---|---|---|---|
| | | | 0 | 10 | 20 | 30 |
| GUW* | Training | Last | 86.67 (0) | 84.58 (1.21) | 82.74 (1.42) | 82.08 (2.50) |
| GDS* | Training | Last | 85.72 (1.65) | 84.63 (1.58) | 84.06 (2.40) | 83.12 (1.91) |
| RAD [22] | Val | Last | **83.89** (0) | 81.80 (0.37) | 0 (0) | 0 (0) |
| SELF [11] | Val | Last | 83.48 (0) | 60.48 (6.09) | 59.99 (4.52) | 60.71 (3.70) |
| END [16] | Val | Full | 82.6 (2) | 79.7 (1) | **81.1** (2) | **77.8** (3) |
| kNN-RAD | Val | Last | **83.89** (0) | **83.20** (1.53) | **82.29** (1.25) | **79.5** (1.91) |
| kNN-SELF | Val | Last | 83.48 (0) | 72.96 (22.96) | 75.77 (8.02) | 73.98 (4.40) |

## 5 Discussion and Limitations

We observe that kNN label spreading dramatically increases the robustness of both RAD and SELF and additionally find several interesting trends that warrant discussion. First, it is clear that for larger noise levels, a larger number of nearest neighbors is needed to correct for the noise. This is suggested by Gao et al. [5] for large $k$, but is empirically verified with our experiments. The empirically optimal $k$ value increases with each increase in label noise, which allows the label spreading to average over a larger number of nearest neighbors. For CMNIST, we must use a smaller, but still increasing, $k$ to account for the smaller clusters within classes. A limitation of our method is that $k$ is chosen as a hyperparameter, but these insights help to suggest likely ranges for the optimal. Indeed, if an estimate of the noise prevalence is available, the choice of $k$ nearest neighbors could reasonably be estimated without hyperparameter tuning at all.

We also note that the quality of the embeddings is crucial in the selection of the number of neighbors and the number of spreading rounds. CMNIST is the prime example of a dataset that is generally well-separated but has not yet experienced "neural collapse," so each class has several clusters of points. The danger of this is that some clusters may be nearer to the opposite class than their own, requiring fewer nearest neighbors to prevent noisy labels from spreading. The base model accuracy can help indicate how extensively it is trained, though embeddings can also be analyzed qualitatively. As pointed out in Section 3, a limitation of our method is that we assume that embeddings are from a model trained on clean data, but this need not be the case. Indeed Iscen et al. [9] use the robustness of embeddings to label noise in order to train robust deep learning models and we verify the robustness of kNN-RAD in particular in Section 4.3.

Table 5: **Waterbirds WGA (std. dev.)**: We see that both kNN-RAD and kNN-SELF strongly outperform END even though they update only the last layer. Non-robust methods fail quickly.

| Method | Domain Annotation | Layer | Label Noise (%) | | | |
| --- | --- | --- | --- | --- | --- | --- |
| | | | 0 | 10 | 20 | 30 |
| GUW* | Training | Last | 91.60 (0.05) | 90.90 (0.78) | 88.25 (1.91) | 84.78 (3.53) |
| GDS* | Training | Last | 92.32 (0.58) | 89.53(1.66) | 86.93 (2.40) | 78.09 (3.36) |
| RAD [22] | Val | Last | 91.23 (0.06) | 79.33 (1.38) | 50.74 (2.23) | 19.52 (1.91) |
| SELF [11] | Val | Last | **92.83** (0.49) | 7.58 (2.77) | 1.38 (0.20) | 0.66 (0.14) |
| END [16] | Val | Full | 82.8 (1) | 84.2 (1) | 83.2 (1) | 81.8 (1) |
| kNN-RAD | Val | Last | 90.92 (0.08) | **90.72** (0.63) | **89.93** (1.10) | **86.90** (3.07) |
| kNN-SELF | Val | Last | **92.65** (0.47) | 89.55 (0.65) | 88.24 (2.93) | 82.44 (10.43) |

Table 6: **Civil Comments WGA (std. dev.)**: SELF struggles on this highly class-imbalanced dataset, but kNN-RAD is competitive with domain-aware methods even for large noise. Oh et al. [16] do not consider this dataset and thus, results for END are not reported.

| Method | Domain Annotation | Layer | Label Noise (%) | | | |
| --- | --- | --- | --- | --- | --- | --- |
| | | | 0 | 10 | 20 | 30 |
| GUW* | Training | Last | 80.25 (0.03) | 80.04 (0.43) | 81.58 (0.18) | 79.39 (0.70) |
| GDS* | Training | Last | 79.67 (0.48) | 80.16(0.63) | 81.18 (0.53) | 81.21 (0.32) |
| RAD [22] | Val | Last | **80.99** (0.03) | 79.25 (0.44) | 77.45 (0.71) | 54.36 (1.30) |
| SELF [11] | Val | Last | 60.61 (0.04) | 59.92 (0.03) | 59.92 (0.04) | 59.95 (0.03) |
| END [16] | Val | Full | - | - | - | - |
| kNN-RAD | Val | Last | **81.0** (0.03) | **81.15** (0.60) | **80.70** (0.62) | **78.56** (1.52) |
| kNN-SELF | Val | Last | 60.65 (0.05) | 72.30 (1.58) | 64.64 (2.72) | 61.30 (0.72) |

Finally, we note that while kNN label spreading works well with both SELF and RAD, the down-sampling to balance the class aspect of the SELF algorithm combined with label spreading leads to a much larger variance than with kNN-RAD. That said, preprocessing with a kNN label spreading module still achieves significant gains in worst-group accuracy relative to vanilla SELF. A coupled label cleaning and error set selection process may be an interesting direction for future work. By leaving our preprocessing generic, we can ensure our method is applicable when future two-stage methods are developed.

Our work allows for the robustness of two-stage fairness corrections, which could improve fairness for minority groups in a wide variety of models. Unfortunately, no two-stage last layer correction can provide guarantees on the fairness of the resulting model in the general setting. This could lead to practitioners assuming fairness without auditing it.

# 6   Acknowledgements

Nathan Stromberg, Rohan Ayyagari, and Lalitha Sankar acknowledge support by NSF CIF-2007688, SCH-2205080, PIPP-2200161, and a Google AI for Social Good grant. Sanmi Koyejo acknowledges support by NSF Career Award 2046795 and SCH-2205329, Stanford HAI, and Google Inc.

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

# A  Experimental Details

The upstream base models for all the datasets which are considered are taken from [22]. This includes all the hyperparameters, data augmentations and training procedures that we use for the upstream models. We use these base models to generate the embeddings for the predefined validation and test splits of the datasets. In our experimental setup, we assume access to a small set of clean data which we use for validation and hyperparameter selection. We split the validation data into two halves. One half is used as the aforementioned clean set for validation and the other half is used for training during hyperparameter selection. During the final testing, we use the entire (noisy) validation split for retraining and the test split for testing. Once we do the hyperparameter selection, we run the algorithms over 10 different noise seeds and report the accuracy and variance over these 10 runs. This procedure is repeated for each dataset and each noise level.

## A.1  Last Layer Retraining Methods

For the methods that involve only last-layer retraining (either with downsampling or upweighting), we only tune c, which is the inverse of the $\lambda$, the $\ell_1$ regularization strength. For all datasets, we tune over 10 values equally spaced on the log scale ranging from 1e-4 to 1.

## A.2  RAD

For each dataset, we fix the c values for both the identification and retraining models. We get these values from [22]. We tune the upweight factor for each dataset and each noise level. The hyperparameter values and ranges used are given in Table 7. The identification model is implemented using the `PyTorch` library and the retraining model is implemented using the `sklearn.linear_model.LogisticRegression` package. The `learning rate (id)` is the learning rate of the identification model and the `epochs (id)` is the number of epochs used to train the identification model.

## A.3  kNN-RAD

We follow the same procedure for kNN-RAD as we did for RAD but with the added kNN based label spreading preprocessing step. In addition to the upweight factor, we also tune `n_neighbors`, which is the number of nearest neighbours used in the kNN algorithm. When there is no noise, we fix the `n_neighbors` value at 1. We use the `sklearn.neighbors.KNeighborsClassifier` package to perform the label spreading step. The ranges and hyperparameter values used are given in Table 8.

## A.4  SELF

We implemented misclassification-SELF using code adapted from LaBonte et al. [11] so that it would be compatible with our setup where we use pre-generated embeddings from the base models. We fix the `finetuning steps` which the number of steps of fine-tuning we perform once we construct the class balanced error set. We tune the learning rate and the number of points that are selected for class balancing. The hyperparameter values and ranges used are given in Table 9

## A.5  kNN-SELF

The procedure for kNN-SELF is the same as SELF but with the added kNN based label spreading preprocessing step. We fix the hyperparameters using the values selected in the hyperparameter selection in SELF. We additionally tune `n_neighbors`. We use the `sklearn.neighbors.KNeighborsClassifier` package to perform the label spreading step. The ranges and hyperparameter values used are given in Table 10

Table 7: **RAD Hyperparameters**

| Dataset | c (id) | c (retraining) | LR (id) | epochs (id) | upweight factor range |
|---|---|---|---|---|---|
| CelebA | 6.16e-4 | 0.007848 | 1e-5 | 6 | [5, 10, 25, 50] |
| Waterbirds | 3.0e-6 | 0.143845 | 1e-5 | 60 | [5, 10, 25, 50] |
| CMNIST | 33.6 | 0.007848 | 1e-5 | 6 | [1, 3, 20, 30] |
| Civilcomments | 6.95e-07 | 0.001833 | 1e-5 | 6 | [1, 3, 6, 10] |

Table 8: **KNN - RAD Hyperparameters**

| Dataset | c (id) | c (retraining) | LR (id) | epochs (id) | num neighbors range | upweight factor range |
|---|---|---|---|---|---|---|
| CelebA | 6.16e-4 | 0.007848 | 1e-5 | 6 | [5, 11, 21] | [10, 25, 50, 75] |
| Waterbirds | 3.0e-6 | 0.143845 | 1e-5 | 60 | [5, 11, 21, 31] | [10, 25, 50, 75] |
| CMNIST | 33.6 | 0.007848 | 1e-5 | 6 | [3, 5, 7] | [1, 3, 20, 30] |
| Civilcomments | 6.95e-07 | 0.001833 | 1e-5 | 6 | [5, 11, 21] | [6, 10, 25, 50] |

Table 9: **SELF Hyperparameters**

| Dataset | fine-tuning steps | learning rate range | num points range |
|---|---|---|---|
| CelebA | 500 | [1e-6, 1e-5, 1e-4] | [2, 20, 100] |
| Waterbirds | 500 | [1e-4, 1e-3, 1e-2] | [20, 100, 500] |
| CMNIST | 500 | [1e-5, 1e-4, 1e-3] | [100, 500, 700] |
| Civilcomments | 200 | [1e-6, 1e-5, 1e-4] | [20, 100, 500] |

Table 10: **KNN - SELF Hyperparameters**

| Dataset | fine-tuning steps | learning rate | num points | num neighbors |
|---|---|---|---|---|
| CelebA | 500 | 1e-5 | 2 | [11, 25, 37] |
| Waterbirds | 500 | 1e-4 | 500 | [5, 11, 21] |
| CMNIST | 250 | 1e-5 | 500 | [7, 21, 41] |
| Civilcomments | 200 | 1e-6 | 500 | [11, 31, 41] |

### A.6 Compute Resources

Experiments in Section 4 and Appendix C were conducted on a supercomputing cluster using NVIDIA GPUs for hardware acceleration. Most compute time is spent training base models which needs to be done just once per dataset. Beyond that, experiments finish within hours.

## B  RAD Fails Under Label Noise

Consider the data distribution introduced in Yao et al. [27] (also used in[26]) with group-conditional Gaussians aligned such that the difference between means within a class is given by $[C_1, 0, \ldots, 0] \in \mathbb{R}^d$ and the difference between means across classes is given by $[0, C_2, 0, \ldots, 0] \in \mathbb{R}^d$. That is, we have axis-aligned core and spurious features. Refer to Figure 3 for a visual.

**Proposition 2** (Axis-Aligned Gaussian Mixture). *Let RAD use the fixed upweighting scheme $\frac{|\mathcal{E}^c|}{|\mathcal{E}|}$. For the dataset described above, with $C_2 > C_1 > C_0$ for sufficiently large $C_0$, RAD fails entirely when learning with symmetric label noise $p > 0$. Otherwise, RAD fails if $\frac{1}{2} > p > 2\pi_0$, achieving lower worst-group accuracy than ERM.*

*Proof.* We assume that $\lambda$, the $\ell_1$ regularization strength of the biased classifier, is large enough that only the first (spurious) or second (core) feature is learned. Consider the two cases separately:

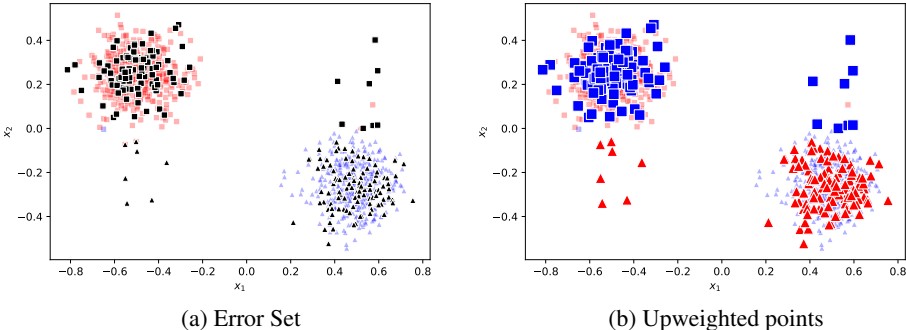

(a) Error Set          (b) Upweighted points

Figure 3: Label noise causes the RAD algorithm to select, and then upweight, the incorrect error set. We show that RAD fails under suffient SLN for imbalanced settings.

**Spurious Feature is Learned**    Consider the error set $\mathcal{E}$. It is made of up clean minority samples and noisy majority samples. When upweighting, the effective noise level in the retraining set will increase for the majority and decrease for the minority. For $2\pi_0 < p < \frac{1}{2}$, as $\pi_0$ goes to 0, the effective noise in the upweighted majority is 50% and 0% in the upweighted minority. In this case, the optimal separator will split the majorities and correctly classify the minorities. By assumption, $C_2 > C_1$ so the $\ell_1$ penalty in the retraining step will force the spurious feature to be learned again. Note that ERM would learn a classifier biased to the majorities, but not be restricted to only the spurious feature. Thus RAD performs worse than ERM.

**Core Feature is Learned**    The error set $\mathcal{E}$ now consists of entirely noisy samples which are upweighted by a factor of $\frac{1-p}{p}$. This results in an effective noise level at retraining of 50%, making the retraining task impossible. Thus RAD fails.       □

## C   $\alpha$-RAD, Why Robust Losses are Not Enough

It may be tempting to believe that simply using a robust loss should be enough to make two-step methods robust to label noise. The issue is that while robust losses can learn a classifier which performs well on clean data, it will (correctly) misclassify noisy examples. This leads to noisy points (which will much more likely to be from a majority group) being selected for the error set which is used to retrain the final classifier. Thus the final classifier will not be trained on data from minority groups as intended, but on mostly majority points, exacerbating unfairness. To demonstrate this, we train RAD [22] using $\alpha$-loss in the identification step. $\alpha$-loss [23] has been demonstrated to be robust to symmetric label noise both theoretically [24] and empirically [24, 23].

We first examine the types of points which are selected by RAD in the error set. In Figure 4 we see that for all noise levels, true minority examples are misclassified (and therefore selected) but as the noise increases, the amount of noisy (former) majority points increases as well. These points drown out the effect of the true minority points, causing failure of RAD at large noise levels. We see in the Table 11 that even when training with this robust loss, worst-group accuracy experiences a dramatic dropoff at larger noise levels. This is the same failure mode that is demonstrated with vanilla cross entropy loss in the main text Section 4.

Table 11: $\alpha$-**RAD**

| Dataset | Label Noise (%) | | | | |
|---|---|---|---|---|---|
| | 0 | 5 | 10 | 15 | 20 |
| CelebA | 79.33 (0.48) | 83.78 (1.28) | 80.67 (1.78) | 58.07 (0.1) | 46.45 (0.15) |
| CMNIST | 91.68 (0.37) | 88.03 (1.47) | 90.56 (1.08) | 80.68 (1.08) | 47.17 (2.56) |

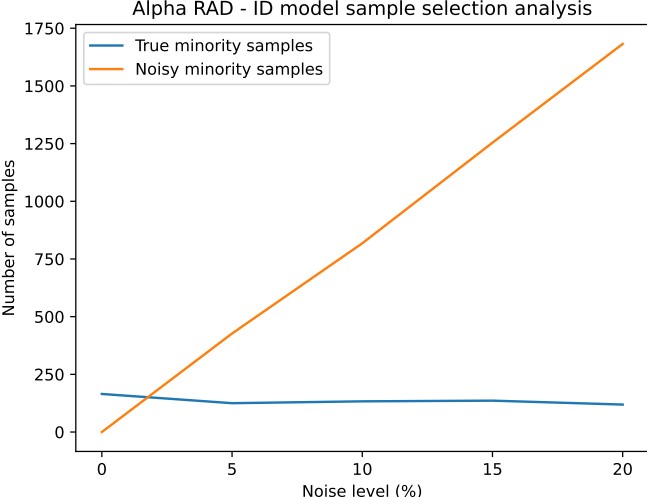

Figure 4: RAD trained with $\alpha$-loss is able to capture minority points at all noise levels, but an increasing number of noisy majority points are selected as noise increases. This leads to poor downstream fairness

