# OpenReview forum: "Enhancing Robustness of Last Layer Two-Stage Fair Model Corrections"
_NeurIPS.cc/2024/Conference — NeurIPS 2024 poster_

### Official Review · Reviewer_phDr · 2024-06-20

**Soundness:** 2
**Presentation:** 3
**Contribution:** 2
**Rating:** 5
**Confidence:** 4

**Summary:**

This paper proposes a simple kNN-based label noise correction strategy to improve the performance of two-stage last-layer retraining methods for group robustness under moderate label noise. The authors show that the performance of RAD and SELF deteriorates quickly when label noise is present in the held-out dataset and show that their kNN method achieves better worst-group accuracy on several benchmark datasets.

**Strengths:**

1. This paper deals with an important problem in group robustness, which is mitigating the impact of class label noise when group annotations are not available. Label noise is especially relevant for methods which use class annotations and model knowledge as a proxy for group annotations (e.g., RAD and SELF).

2. The proposed kNN method is post-hoc and training-free, which means computational cost is negligible and it can easily be added to existing training pipelines.

3. Benchmark evaluation is comprehensive, covering 4 well-known datasets and recent competitive methods, and using means and standard deviations over 10 seeds.

**Weaknesses:**

1. I encourage the authors to discuss any “baseline” label noise in the given datasets; for instance, Waterbirds is known to contain incorrect labels [1] and therefore the standard dataset has non-zero label noise. This suggests that methods like RAD and SELF might already be robust to a small amount of label noise.

2. I am confused by the assumptions in Section 2. It is stated that $f$ is a neural network pretrained on clean data, whose last layer is then retrained to obtain the final model. What exactly is this pretraining dataset? Is it the weight initialization (e.g. ImageNet) or the downstream dataset used for ERM finetuning (e.g. Waterbirds)? If it is the former, I believe the assumption is justified, but the training procedure should be clarified to include a round of ERM finetuning. If it is the latter, I believe the assumption is not justified, as the held-out set for LLR is essentially a random subset of the finetuning dataset and should therefore obey the same label distribution. Indeed, if the finetuning dataset was clean but the held-out dataset was noisy, one should just throw out the held-out dataset and hold out a subset of the finetuning dataset instead.

3. The authors focus on the “misclassification” version of the SELF algorithm. However, [2] shows that misclassification is actually the worst version of SELF, particularly on CivilComments, and they ultimately propose early-stop disagreement SELF which has much better performance. What is the justification for using the misclassification version of SELF when its initial performance is so low, and why is disagreement SELF not used? I wonder if this would also help with the variance problem of kNN-SELF noted in Section 5, as [2] show reduced variance using the disagreement technique as well.

***References***

[1] Taghanaki et al. “MaskTune: Mitigating Spurious Correlations by Forcing to Explore”. NeurIPS 2022.

[2] LaBonte et al. “Towards Last-layer Retraining for Group Robustness with Fewer Annotations”. NeurIPS 2023.

**Questions:**

1. Should lowercase $x_j$ be used in Section 3 instead of uppercase? I thought uppercase $X$ represented the entire dataset while lowercase $x_j$ was a single point.
2. Are group annotations necessary for hyperparameter tuning of the kNN methods? This is an important limitation and should be made more explicit.
3. Regarding Weakness #2, what is the performance of the proposed method if the finetuning dataset is noisy?

**Limitations:**

The limitations and social impact of the proposed kNN method are sufficiently discussed, but my concerns from the Weaknesses section may constitute important limitations if not addressed.

---

> ### Author Rebuttal · Authors · 2024-08-06
>
> We thank the reviewer for their thoughtful comments, especially their acknowledgment of the strength of our evaluation and method in general. We would like to address each question (Qx) and weakness (Wx) individually. Note that our references continue numbering from the review.
>
>  (W1)	Regarding the baseline level of noise in the data, while it may be true that these datasets have inherent noise, the noise is consistent across all data splits, meaning that there is no distribution shift from training time to test time. Therefore RAD and SELF are likely not “robust,” they just fit to the distribution they are trained on. Further discussion about the limitations of current benchmarks is always warranted, though, and we thank the reviewer for their comment.
>
>  (W2/Q3)	To clarify the noise model, we currently make no assumptions about the pretraining set (e.g., ImageNet) and assume that the full-finetuning dataset (e.g., training split of CelebA) is clean. While this assumption may seem strong, as we pointed out in section 3 of our paper and the discussion section, Iscen, et al. [3] find that embeddings are generally fairly robust to label noise, and only the final classification layer is strongly affected by noise. Furthermore, all last-layer retraining methods should be affected similarly by the quality of the embeddings. Still, we think it is important to examine this assumption more closely, and have run preliminary experiments to demonstrate that our method is robust to the violation of this assumption. Full results are shared in Tables 1-2 in the General rebuttal. We full finetune models for both CelebA and Waterbirds with 20% label noise and then use these embeddings to test the LLR methods. We see that both RAD and SELF perform very poorly in this scenario, though RAD seems much more resilient to poor-quality embeddings. As the noise increases in the finetuning set, RAD and SELF both decline quickly in WGA. Utilizing kNN label spreading, however, provides much-improved robustness, on par or better than gains we see in the clean-embeddings experiments. Thus, we conclude that our method is robust to violations (even major ones) of our “clean embeddings” assumption.
>
>  (W3)	We focus on the misclassification variant of SELF because it requires the least side information about the unfair base model. While ES-SELF performs well on CivilComments, it requires access (as suggested in [2] and confirmed in their codebase) to early-stopped versions of the base-model to which we do not assume access. Additionally, there is no reason to believe that disagreement should solve the problem of noisy labels. Indeed, noisy points may be unnecessarily included in the error set for upweighting.
>
>  (Q1)	Regarding the capitalization of $X$ in section 3, we intend the use of capitals to denote random variables and use this notation when discussing theory. For the algorithms, we use lowercase x to denote that we have been given a realization of the random variable $X$. We will ensure that this is clarified in the camera-ready version of the paper.
>
>  (Q2)	Currently, the hyperparameter is selected using the clean, labeled validation set, but this requirement is not firm. Indeed, in our testing, we see strong agreement between the target predictive power of kNN on validation data and WGA on clean validation data (that is, the better we recover target labels, the better our downstream WGA is). Thus, while we currently use domain annotations to tune $k$, there is promise in proxy methods for tuning this important hyperparameter. We will make this point clear in the camera-ready version of the paper.
>
> We hope that this rebuttal helps to assuage your concerns and highlights the important contributions of our method.
>
> **References**
>
> [3] A. Iscen, J. Valmadre, A. Arnab, and C. Schmid. Learning with neighbor consistency for noisy labels. In Proceedings of the IEEE/CVF Conference on Computer Vision and Pattern Recognition, pages 4672–4681, 2022

---

> ### Comment · Reviewer_phDr · 2024-08-09
>
> Thanks to the authors for their comprehensive response.
>
> For (W2), my concerns have been partially addressed, especially with the good performance of the kNN method using noisy finetuning datasets -- though I still have reservations about the practicality of the clean finetuning dataset assumption, since in practice one could just perform LLR/RAD/SELF using a held-out subset of the clean dataset. I look forward to reading more comprehensive experiments on noisy finetuning datasets in the final version.
>
> For (W3), the authors bring up an interesting point about whether disagreement can solve the problem of noisy labels. In [2] the authors provide some discussion on this matter, as they claim misclassification selects "difficult" data (more likely to be noisy) while disagreement selects "uncertain" data (more likely to be legitimate minority group data). While somewhat orthogonal to the proposed kNN method, it would be valuable to the community to provide some discussion of this point, perhaps including experiments showing whether misclassification or disagreement methods are better at filtering noisy labels.
>
> Regardless, though, I think some comparison to early-stop disagreement SELF should be included (and the authors can justify in the text that the numbers are not directly comparable due to the additional information of an early-stopped model), since [2] is clear that misclassification SELF is not the recommended method. In fact, they get much better performance on CivilComments by training on a *random* subset of data.
>
> For (Q2), I suggest that the authors include model selection using worst-class accuracy [4] or the bias-unsupervised validation score [5] in the final version. These are recent proposals for proxy methods that do not use group annotations; they seem to perform well and are not difficult to implement.
>
> **Recommendation**
>
> Overall, this paper is borderline for me. The problem is important and the proposed method is interesting, but I have remaining concerns about the assumptions, evaluations, and comparisons as discussed above. With that said, I now lean slightly towards acceptance instead of rejection, so I have raised my score to a 5.
>
> **References**
>
> [2] LaBonte et al. “Towards Last-layer Retraining for Group Robustness with Fewer Annotations”. NeurIPS 2023.
>
> [4] Yang et al. "Change is Hard: A Closer Look at Subpopulation Shift". ICML 2023.
>
> [5] Tsirigotis et al. "Group Robust Classification Without Any Group Information". NeurIPS 2023.

---

### Official Review · Reviewer_e2kD · 2024-07-12

**Soundness:** 4
**Presentation:** 4
**Contribution:** 2
**Rating:** 6
**Confidence:** 4

**Summary:**

This paper addresses the challenge of improving worst-group accuracy (WGA) in machine learning models, particularly in the presence of noisy labels. The authors focus on last-layer retraining (LLR) methods, which have emerged as an efficient approach for correcting existing base models to ensure fairness across subgroups.

The key contributions of the paper are:

1. Highlighting the vulnerability of state-of-the-art LLR methods, specifically SELF and RAD, to label noise in the training data.

2. Introducing a novel label correction preprocessing method based on k-nearest neighbors (kNN) label spreading. This method significantly improves the performance of LLR methods under label noise conditions.

3. Proposing two new algorithms, kNN-RAD and kNN-SELF, which combine the kNN label spreading technique with existing LLR methods (RAD and SELF, respectively).

4. Demonstrating the effectiveness of their approach across various spurious correlation datasets, including CMNIST, CelebA, Waterbirds, and CivilComments.

The authors show that their proposed methods, particularly kNN-RAD, achieve state-of-the-art performance in terms of worst-group accuracy without requiring domain annotations during training. The approach is competitive with, and in some cases outperforms, domain-aware methods and full model retraining approaches like END.

The paper provides both theoretical insights and empirical evidence for the effectiveness of kNN label spreading in correcting noisy labels. It also discusses the relationship between the optimal number of nearest neighbors and the level of label noise, offering practical guidance for implementing the method.

Overall, this work presents a simple yet effective approach to enhancing the robustness of last-layer retraining methods for improving worst-group accuracy in the presence of label noise, addressing an important challenge in the field of fair machine learning.

**Strengths:**

### Originality:
1. It creatively combines existing ideas from label propagation and worst-group accuracy correction, applying them in a novel context to address label noise in fairness-oriented model corrections.
2. The authors introduce a new problem formulation by focusing on the robustness of last-layer retraining methods to label noise, an issue that had not been thoroughly addressed in previous work.
3. The proposed kNN label spreading preprocessing step is an innovative approach to improving the robustness of existing methods like RAD and SELF without fundamentally altering their core algorithms.

Quality:
1. The authors provide both theoretical insights (e.g., the relationship between optimal k and noise level) and extensive empirical evidence to support their claims.
2. The experiments are comprehensive, covering multiple datasets (CMNIST, CelebA, Waterbirds, CivilComments) and comparing against state-of-the-art methods as well as oracle baselines.
3. The paper includes detailed ablation studies, examining the effects of different numbers of neighbors and spreading rounds, which adds depth to the analysis.
4. The authors are transparent about limitations and potential issues, such as the dependence on well-separated embeddings.

Clarity:
The paper is well-structured and clearly written:
1. The problem setup and background are concisely explained, making the work accessible to readers familiar with machine learning concepts.
2. Algorithms are presented in pseudocode, enhancing reproducibility.
3. Results are presented in well-organized tables with clear explanations of the experimental setup and findings.
4. The discussion section provides insightful analysis of the results and addresses potential limitations.

Significance:
1. It addresses an important problem in fair machine learning – improving worst-group accuracy in the presence of noisy labels – which is crucial for real-world deployments of AI systems.
2. The proposed method is computationally efficient (last-layer retraining) and does not require domain annotations, making it widely applicable in various settings.
3. The strong performance across different datasets suggests broad applicability of the approach.
4. By improving the robustness of existing methods, this work potentially extends the usability of fairness-correcting algorithms in more challenging, real-world scenarios.
5. The insights provided about the relationship between noise levels and optimal nearest neighbors could guide future research in this area.

**Weaknesses:**

While the paper presents valuable contributions, there are some areas where it could be improved:

1. Limited theoretical analysis:
   The paper provides some theoretical insights, particularly referencing Gao et al.'s work on kNN classification with label noise. However, a more rigorous theoretical analysis specific to this method could strengthen the paper. For instance:
   - A formal proof of convergence for the label spreading algorithm in this context.
   - Theoretical bounds on the expected improvement in worst-group accuracy after applying kNN label spreading.
   - An analysis of how the method's performance depends on the separation of classes in the embedding space.

2. Sensitivity to hyperparameters:
   The authors acknowledge that the choice of k (number of nearest neighbors) is crucial and depends on the noise level. While they provide some empirical guidance, a more systematic approach to selecting k would be beneficial. For example:
   - A heuristic method for estimating the optimal k based on dataset characteristics and estimated noise level.
   - An analysis of the method's sensitivity to suboptimal choices of k.

3. Assumptions about embedding quality:
   The method relies heavily on the quality of the embeddings from the base model. While the authors discuss this limitation, they could expand on:
   - How the method performs when the base model is trained on noisy data, violating the clean data assumption.
   - Potential approaches to improve embedding quality in the presence of noisy labels.

4. Limited exploration of other label propagation techniques:
   The paper focuses on kNN label spreading, but other label propagation techniques exist. A brief comparison with alternative methods (e.g., graph-based label propagation) could provide more context for the choice of kNN.

5. Comparison with other robust learning methods:
   While the paper compares with END and domain-aware methods, a comparison with other robust learning techniques (e.g., importance reweighting, robust loss functions) could provide more context for the method's effectiveness.

**Questions:**

1. Robustness to noisy embeddings:
   Question: How does the performance of kNN-RAD and kNN-SELF change when the base model is trained on noisy data, violating the clean data assumption for embeddings?

2. Optimal selection of k nearest neighbors:
   Question: Given that the optimal number of nearest neighbors ($k$) depends on the noise level, which is often unknown in practice, how can practitioners best select this crucial hyperparameter?

3. Extension to multi-class problems:
   Question: How does the proposed method extend to multi-class classification problems, and what additional challenges might arise in this setting?

**Limitations:**

Authors have discussed the limitations sufficiently.

---

> ### Author Rebuttal · Authors · 2024-08-06
>
> We appreciate the reviewer’s thorough analysis of our submission and hope that we can answer some of the questions presented. We answer each question (Qx) and weakness (Wx):
>
>  (Q1/W3) 	Regarding the need for clean embeddings, as we point out in the discussion and in section 3 of our submission, Iscen, et al. [1]  explicitly exploit the robustness of these embeddings to label noise in order to detect and correct outliers. To assess the downstream effect of noise in the embedding, we test our method on CelebA and Waterbirds using embeddings learned with label noise. We see that both RAD and SELF perform very poorly in this scenario, though RAD seems much more resilient to poor quality embeddings. As the noise increases in the finetuning set, RAD and SELF both decline quickly in WGA. Utilizing kNN label spreading, however, provides much improved robustness. Thus we conclude that our method is robust to violations of our “clean embeddings” assumption.
>
>  (Q2/W2)	Regarding the selection of k nearest neighbors, it is possible to estimate the noise parameter from data as suggested in [2], but in practice k is relatively easy to tune through cross validation. On CelebA, we examine the performance of kNN-RAD with suboptimal k and see that selecting k larger than optimal results in a steady decline in WGA, but selecting k too small yields much worse downstream performance. This suggests that erring on the side of large k (10-20 in our tests) is generally better than too small. Even suboptimal performance (too large of k) results in vastly increased robustness over vanilla RAD (0% WGA). Full results are in Table 3 of the general rebuttal.
>
>  (Q3)	The problem of label propagation in the multi-class setting is well studied, and the implicit geometry of multi-class classifiers is still amenable to propagation in the latent space. The noise model can be more complex in the multi-class setting, but if the noise flips to other classes completely at random, then our proposed method will still be effective. Additionally, both RAD and SELF have been shown to be effective in the multi-class setting. Due to time limitations, we do not have results on a multi-class dataset, but this certainly a good direction to explore.
>
>  (W1)	Regarding additional theoretical analysis, our method is inspired by prior theoretical work which proves the robustness of kNN to label noise and combining this with empirically effective, but fragile, two-stage correction methods. We carry over the guarantees of robustness for kNN, but work remains to be done to understand the theoretical guarantees of two-stage corrections on their own. We are currently pursuing this direction, and we believe theoretical analysis for the interplay between kNN and two-stage corrections is promising future work. Experiments suggest that not only the separation of the classes makes a difference as suggested, but also the separation of subgroups within classes.
>
>  (W4)	We appreciate the reviewer’s suggestion of additional label propagation techniques and believe that this could be an interesting path moving forward. Label propagation on the kNN graph is appealing because of its relative simplicity and the relationship between kNN and the downstream linear classification task in RAD and SELF. Indeed, as the number of nearest neighbors grows, the classification boundary becomes more and more (locally) linear, which in turn aids robustness of the downstream LLR model learned for fairness correction. On some datasets (e.g., CMNIST), this linearity induced by large k may be undesirable, and another graph structure or label propagation method may be more appropriate. The joint design of fairness correction and label correction in this manner is a compelling area of future research.
>
>  (W5)	Regarding comparison to robust loss functions, in our preliminary exploration, we believed that these methods may have promise, but our experimental results were poor. Fundamentally, the objective of a robust loss is to learn a classifier on noisy data that predicts well on clean data. This means that noisy data is (correctly) misclassified by these models and, thus, if used in the first stage of RAD or SELF, would promote the inclusion of these noisy points into the error set. This, in turn, assigns more weight to noisy examples in the final retraining step, thereby dramatically reducing performance and fairness. Alternatively, label propagation corrects these labels explicitly before training, which prevents noisy points from dominating the error set.
>
> We hope we have adequately addressed the presented concerns and demonstrated the significance of our contribution.
>
> **References**
>
> [1] A. Iscen, J. Valmadre, A. Arnab, and C. Schmid. Learning with neighbor consistency for noisy labels. In Proceedings of the IEEE/CVF Conference on Computer Vision and Pattern Recognition, pages 4672–4681, 2022
>
> [2] Patrini, G., Rozza, A., Krishna Menon, A., Nock, R., and Qu, L. Making deep neural networks robust to label noise: A loss correction approach. In Proceedings of the IEEE conference on computer vision and pattern recognition, 2017.

---

> > ### Comment · Reviewer_e2kD · 2024-08-08
> > **Thanks for your response**
> >
> > Thank you for your detailed response. I think it generally addresses my concerns and answers my questions.

---

> > > ### Author Response · Authors · 2024-08-08
> > >
> > > We are glad that we could address your concerns. We believe that the incorporation of these points into our paper significantly strengthens our submission, and we hope that you consider increasing your score accordingly.

---

### Official Review · Reviewer_9x3j · 2024-07-14

**Soundness:** 3
**Presentation:** 4
**Contribution:** 2
**Rating:** 6
**Confidence:** 5

**Summary:**

This paper examines the recently introduced last-layer retraining (LLR) method, which focuses on reweighting features to ensure fairness and improve worst-group performance with minimal group annotation. The authors point out the shortcomings of the LLR method, particularly when label noise is present. To address this, they propose a label correction method using label propagation, assuming that LLR operates on largely separable embeddings. This new approach enhances the performance of state-of-the-art LLR methods under label noise.

**Strengths:**

- The proposed method effectively combines existing works.
- Shows very strong performance.
- The  paper is very well-written and easy to follow.

**Weaknesses:**

- The work relies on assumptions such as the need for clean data and the availability of high-quality embeddings.
- Lack of originality.
- Missing recent work [1] which identifies minority groups without using an early-stopped model, instead relying on unsupervised object-centric concept discovery.

**Questions:**

- The authors assume the need for clean data, which I believe is a strong assumption for real-world applications. How do RAD and SELF perform when using only the clean data (set aside for validating the proposed method), given that these methods are designed to be sample-efficient alternatives?
- Can the authors show effectiveness of the proposed method in multi-class and multi-bias dataset [2]?

References

[1] Arefin, Md Rifat, et al. "Unsupervised Concept Discovery Mitigates Spurious Correlations." Forty-first International Conference on Machine Learning, 2024.

[2] Li, Zhiheng, et al. "A whac-a-mole dilemma: Shortcuts come in multiples where mitigating one amplifies others." Proceedings of the IEEE/CVF Conference on Computer Vision and Pattern Recognition. 2023.

**Limitations:**

Limitations of the work is discussed well in the paper.

---

> ### Author Rebuttal · Authors · 2024-08-06
>
> We appreciate the reviewer’s considered response to our submission. We would like to politely push back on a few points and answer the reviewer’s questions (Qx) and weaknesses (Wx) in turn. Note that our references continue numbering from the review.
>
>  (Q1/W1)	Regarding the need for clean embeddings, as we point out in the discussion and in section 3 of our submission, Iscen, et al. [3]  explicitly exploit the robustness of these embeddings to label noise in order to detect and correct outliers. To assess the downstream effect of noise in the embedding, we test our method on CelebA and Waterbirds using embeddings learned with 20% label noise. See Tables 1-2 in the general response for full results. We see that both RAD and SELF perform very poorly in this scenario, though RAD seems much more resilient to poor quality embeddings. As the noise increases in the finetuning set, RAD and SELF both decline quickly in WGA. Utilizing kNN label spreading, however, provides much improved robustness. Thus we conclude that our method is robust to violations of our “clean embeddings” assumption.
>
>  (Q1)	Regarding the performance of the RAD and SELF using only the clean holdout, we thank the reviewer for their suggestion of a new baseline. While RAD and SELF are generally sample efficient, SELF specifically downsamples the error set and so will likely suffer more in this setting. Regardless, we have run a preliminary experiment in this direction: we restrict our finetuning set to the clean holdout (used for validating our method) and train RAD and SELF in the usual manner. On CelebA, RAD achieves 74 $\pm$ 9.92 and SELF achieves 80.89 $\pm$ 0. This is beaten by kNN-RAD up to 30% SLN, suggesting that there is more information to be gained by utilizing the noisy embeddings. On Waterbirds, RAD achieves 83.4 $\pm$ 6.35 and SELF achieves 60.57 $\pm$ 12.29. Here using a kNN method is superior at every tested noise level. These tests use half the available data (we assume the validation split is clean), but this could conceivably be reduced by different hyperparameter tuning strategies. In that case, we expect the gap between kNN-corrected methods and vanilla two-stage methods to grow.
>
>  (W3)	We appreciate the reviewer’s suggestion of [1], a very recent work on unsupervised concept discovery. While their method has the advantage of reduced reliance on holdout annotations, it appears significantly more computationally intensive, involving training two embedding networks to identify concepts. The aim of our paper is to leverage existing embeddings to train a fair classifier with limited data, thus increasing the reuse of powerful pre-trained models. We believe, however, that these concepts could be used to improve future last layer retraining (LLR) methods.
>
>  (Q2)	The multi-class, multi-bias setting is a very interesting one and how well shortcuts (unintended decision rules that are unable to generalize, i.e., spurious features) can be mitigated with access only to the LLR tools without access to jointly optimizing the embeddings is not fully clear. For LLR-only methods, indeed, approaches such as DFR [4] and other related LLR methods explicitly optimize to reduce the reliance of the pretrained model on multiple shortcuts which have already been learned by the embeddings. Unfortunately, due to time constraints we are not able to run experiments with these additional datasets, but these datasets present meaningful benchmarks for future investigation.
>
> We once again thank the reviewer for many valuable suggestions, and we hope that we were able to address the reviewer’s concerns sufficiently and demonstrate the valuable contribution of our method, especially when clean data is limited.
>
> **References**
>
> [3] A. Iscen, J. Valmadre, A. Arnab, and C. Schmid. Learning with neighbor consistency for noisy labels. In Proceedings of the IEEE/CVF Conference on Computer Vision and Pattern Recognition, pages 4672–4681, 2022
>
> [4] P. Kirichenko, P. Izmailov, and A. G. Wilson. Last layer re-training is sufficient for robustness to spurious correlations. In The Eleventh International Conference on Learning Representations, 2023.

---

> ### Author Response · Authors · 2024-08-13
>
> We are very grateful to the reviewer for their valuable suggestions, especially regarding new baselines. We believe that we were able to address the reviewer’s concerns and hope that it warrants an increase in score.

---

> ### Comment · Reviewer_9x3j · 2024-08-13
>
> Thank you to the authors for addressing my questions. The authors presented preliminary experiments regarding my concern about the clean data assumption, where they claim their method shows improvement over RAD and SELF. However, it is unclear why RAD and SELF perform differently on CelebA and Waterbirds, and the significance of their improvement when other methods are trained on similarly clean data. I recommend finishing these experiments and including them in the paper, along with a detailed discussion of the sample complexity of each experiment. Considering the importance of the problem studied and the promising results, I have decided to increase my score.

---

> > ### Author Response · Authors · 2024-08-13
> >
> > Thank you for your response, we will be sure to include the complete set of experiments and discussion in the final version of the paper. Thank you again for your suggestions.

---

### Author Rebuttal · Authors · 2024-08-06

We thank all the reviewers for their insightful comments, and are grateful that they found our work well-written and sound. We would like to address the most common concerns in a general comment and we hope that this demonstrates the strong, and sometimes unexpected, contribution of our method.
 - **Assumption of clean embeddings**:  we can now show through additional experiments that our proposed method is robust to the violation of this assumption, even with large amounts of noise in the training of the embeddings. We ran such experiments for CelebA and Waterbirds datasets. Indeed, in Tables 1-2, we see that kNN label spreading combined with two-stage LLR correction shows massive improvements in WGA over two-stage corrections alone on both CelebA and Waterbirds when the embedding model is learned on noisy data. This demonstrates that the clean embeddings assumption can be weakened without a major hit to performance over vanilla methods.
 - **Use of a clean holdout dataset**: we ran additional experiments to understand if it suffices to train on a smaller clean holdout set using two-stage methods such as RAD and SELF. Our results are in line with our expectation that when using only this clean holdout to train vanilla methods as suggested by Reviewer 9x3j, RAD and SELF have decreased WGA due to the limited amount of training data. On the other hand, if we were to use the larger noisy training dataset, kNN label spreading allows these methods to train on more data and provides downstream WGA benefits even for very large amounts of noise. This is a perfect example of the utility of our method in real-world scenarios where clean data may be limited. In summary, these results validate the original hypothesi of the paper that the use of noisy training data meaningfully increases the downstream performance of two-stage correction methods when cleaned using kNN label spreading.
 - **Selection of optimal $k$** : To better understand the robustness of our proposed method to suboptimal choices of k, we examine the downstream WGA of two-stage methods after kNN label spreading. We see in Table 3 that choosing k too large has a detrimental effect on downstream WGA, but the effect is relatively minor. Choosing k significantly too small can result in a failure as pointed out in Figure 1 of the main paper, but this is easily corrected by erring on the side of large k. This aligns with the idea that classes should be nearly linearly separable in the latent space.
 - **Novelty of our method**: The key novelty of our work is exploiting the inherent structure of the latent space of deep neural networks to efficiently correct for label noise, thereby dramatically increasing the robustness of increasingly popular two-stage fairness corrections. Our results demonstrate that, although our method is lightweight in both data and compute requirements, kNN-RAD and kNN-SELF are significantly more robust to label noise than their vanilla counterparts.

We will ensure that these changes are made for the camera-ready version should the manuscript be accepted.

---

### Decision · Program_Chairs · 2024-09-25

**Decision:**

Accept (poster)

**Comment:**

This paper proposes a robust method for last layer fair model corrections, which tackles the problem of worst group accuracy when the labels are noisy.  Their method uses label spreading from a latent NN graph as a corrective measure on top of last layer fairness correction.  The paper addresses an important problem in fair ML, and constitutes an advance in one of the few effective fairness corrections.  The authors do a good job of highlighting the vulnerability of existing methods.  The reviewers note the strength of the theoretical justification of the method, and the empirical comparisons (which include ablation studies).  Finally, the authors address several criticisms in a revision, particularly regarding the optimal selection of k.  A few weaknesses were pointed out, including limited comparison to other label propagation techniques, and the potentially restrictiveness of the assumptions.